# A new air shower array in the Southern Hemisphere looking for the origins of Cosmic rays: the ALPACA experiment

M. Anzorena[1*], E. de la Fuente[2], K. Fujita[1], R. Garcia[1], K. Goto[3], Y. Hayashi[4],
K. Hibino[5], N. Hotta[6], G. Imaizumi[1] A. Jimenez-Meza[7], Y. Katayose[8], C. Kato[9], S. Kato[1],
T. Kawashima[1], K. Kawata[1], T. Koi[10], H. Kojima[11], T. Makishima[8], Y. Masuda[9],
S. Matsuhashi[8], M. Matsumoto[9], R. Mayta[12,13], P. Miranda[14], A. Mizuno[1], K. Munakata[9],
Y. Nakamura[1], M. Nishizawa[15], Y. Noguchi[8], S. Ogio[1], M. Ohnishi[1], S. Okukawa[8],
A. Oshima[3,10], M. Raljevich[14], H. Rivera[14], T. Saito[16], T. Sako[1], T. K. Sako[17],
T. Shibasaki[17], S. Shibata[11], A. Shiomi[18], M. A. Subieta Vasquez[14], F. Sugimoto[1],
N. Tajima[19], W. Takano[5], M. Takita[1], Y. Tameda[20], K. Tanaka[21], R. Ticona[14],
I. Toledano-Juarez[22], H. Tsuchiya[23], Y. Tsunesada[12,13], S. Udo[5], R. Usui[8], G. Yamagashi[8],
K. Yamazaki[10] and Y. Yokoe[1]

**1** Institute for Cosmic Ray Research, University of Tokyo, Kashiwa 277-8582, Japan.
**2** Departamento de Física, CUCEI, Universidad de Guadalajara, Guadalajara, México.
**3** College of Engineering, Chubu University, Kasugai 487-8501, Japan.
**4** Department of Science and Technology, Shinshu University, Matsumoto 390-8621, Japan.
**5** Faculty of Engineering, Kanagawa University, Yokohama 221-8686, Japan.
**6** Faculty of Education, Utsunomiya University, Utsunomiya 321-8505, Japan.
**7** Departamento de Tecnologías de la Información, CUCEA, Universidad de Guadalajara, Zapopan, México.
**8** Faculty of Engineering, Yokohama National University, Yokohama 240-8501, Japan.
**9** Department of Physics, Shinshu University, Matsumoto 390-8621, Japan.
**10** College of Engineering, Chubu University, Kasugai 487-8501, Japan.
**11** Chubu Innovative Astronomical Observatory, Chubu University, Kasugai 487-8501, Japan.
**12** Graduate School of Science, Osaka Metropolitan University, Osaka 558-8585, Japan.
**13** Nambu Yoichiro Institute for Theoretical and Experimental Physics, Osaka Metropolitan University, Osaka 558-8585, Japan.
**14** Instituto de Investigaciones Físicas, Universidad Mayor de San Andrés, La Paz 8635, Bolivia.
**15** National Institute of Informatics, Tokyo 101-8430, Japan.
**16** Tokyo Metropolitan College of Industrial Technology, Tokyo 116-8523, Japan.
**17** Department of Information and Electronics, Nagano Prefectural Institute of Technology, Nagano 386-1211, Japan
**18** College of Industrial Technology, Nihon University, Narashino 275-8575, Japan.
**19** RIKEN, Wako 351-0198, Japan.
**20** Faculty of Engineering, Osaka Electro-Communication University, Neyagawa 572-8530, Japan.
**21** Graduate School of Information Sciences, Hiroshima City University, Hiroshima 731-3194, Japan.
**22** Doctorado en Ciencias Físicas, CUCEI, Universidad de Guadalajara, Guadalajara, México.
**23** Japan Atomic Energy Agency, Tokai-mura 319-1195, Japan.

\* anzorena@icrr.u-tokyo.ac.jp ,

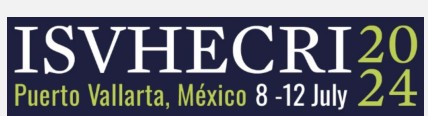

*22nd International Symposium on Very High Energy Cosmic Ray Interactions (ISVHECRI 2024) Puerto Vallarta, Mexico, 8-12 July 2024*
doi:10.21468/SciPostPhysProc.?

## Abstract

The Tibet AS$\gamma$ experiment successfully detected sub-PeV $\gamma$-rays from the Crab nebula using a Surface Array and underground muon detector. Considering this, we are building in Bolivia a new experiment to explore the Southern Hemisphere, looking for the origins of cosmic rays in our Galaxy. The name of this project is Andes Large area PArticle detector for Cosmic ray physics and Astronomy (ALPACA). A prototype array called ALPAQUITA, with 1/4 the total area of the full ALPACA, started observations in September 2022. In this paper we introduce the status of ALPAQUITA and the plans to extend the array. We also report the results of the observation of the moon shadow in cosmic rays.

Received Date
Accepted Date
Published Date

## Contents

## 1 Introduction

The southern celestial hemisphere is know to be rich in astrophysical objects emitting $>100$ TeV $\gamma$-rays [1], which in turn are concentrated near the galactic plane, particularly on the Galactic center. This comes as a result of the $\gamma$-ray attenuation with the CMB photons, hindering the possibility of observing these energetic sources in the current northern observatories. Then, the Southern hemisphere is an excellent location for searching the origin of Galactic Cosmic-rays, since their acceleration could imply the production of $>100$ TeV $\gamma$-rays coming from $\pi^0$ decays. However, sub-PeV photons could also be emitted through the inverse Compton process from sub-PeV electrons, making the solution of this mystery very complex.

The detection and spectral measurement of $>100$ TeV $\gamma$ rays from their celestial sources, together with multi-wavelength (radio and X-ray) observations, will bring key information, allowing us to discriminate between two processes (cosmic-ray/electron origins), locate the acceleration site of cosmic rays (PeVatrons which accelerate cosmic rays up to PeV energies) and to verify the standard acceleration model. The $\gamma$-ray air shower experiments such as the Tibet AS$\gamma$, HAWC and LHAASO, currently in operation in the northern hemisphere, have good sensitivity above 100 TeV and have surveyed the northern sky region with a wide field of view; discovering many sub-PeV gamma-ray sources [2–6]. The Tibet AS$\gamma$ experiment,

has successfully observed gamma rays above 100 TeV from an astrophysical source [2], and diffuse gamma rays >400 TeV from the Galactic plane [7] using a hybrid technique (surface array+underground muon detector).

With this motivation, the Andes Large area PArticle detector for Cosmic ray physics and Astronomy (ALPACA) is a new air shower array project in collaboration between Bolivia, Japan and Mexico. The detector is currently under construction near the Chacaltaya mountain and will be the first experiment to observe high energy $\gamma$-rays in the southern hemisphere. In 2022 the prototype of ALPACA (ALPAQUITA) started observations with around 1/4 of the total size of the full array. In 2025 the construction of the first underground muon detector (MD) and a further extension of the surface array is planed. In this paper, we present the design and the construction plan of ALPACA and ALPAQUITA, then demonstrate the initial performance of ALPAQUITA.

## 2   The ALPACA experiment

ALPACA is aiming at being the first experiment to observe >100 TeV $\gamma$ rays in the Southern Hemisphere. Located at the Chacaltaya plateau in Bolivia ($16°23'$S, $68°8'$W) at an altitude of 4740 m (572 g cm$^{-2}$), ALPACA will observe $\gamma$-ray initiated showers near their maximum development. The experiment is composed of surface air shower (AS) array and underground water Cherenkov muon detectors (MD). MD detectors allow to discriminate the large CR background [8]. The AS array will cover and area of 83 000 m$^2$ with 401 plastic scintillators of 1 m × 1 m × 0.05 m each. The MD array has 4 underground MD pools, each divided in 16 cells of 7.5 m × 7.5 m. The cells in the pools are cover with water proof material to collect the light generated by high energy muons entering the pool (>1.2 GeV). Meanwhile, the 2 m soil overburden works to absorb the electromagnetic component associated with $\gamma$-ray showers. The collected light is detected using a 20-inch PMT installed at the ceiling of the cell. Figure 2 shows a cross section of one MD pool.

In 2022, the prototype of ALPACA, ALPAQUITA, started observations. ALPAQUITA is roughly 1/4 of the full ALPACA, covering an area of 18 450 m$^2$. Figure 1 shows an schematic diagram of the ALPAQUITA. The construction of the underground MDs will start soon. A detailed Monte Carlo (MC) simulation study and performance study of ALPAQUITA, including one MD pool at the center of the array, is reported in [9]. According to this study, ALPAQUITA will be capable of detecting five $\gamma$-ray sources above 10 TeV with high statistical significance ($> 5\sigma$) in one year of observation.

The performance of ALPACA/ALPAQUITA in detecting point-sources is evaluated through MC simulation of the response of the detector to an air shower event (CORSIKA+GEANT4). In this analysis, we reconstruct the events that trigger the detector and then obtain the energy and angular resolution of the experiment. A very important part of this study is the $\gamma$-ray/CR discrimination, which is done by using the information from the MDs. Figure 3 shows the output from MC simulation comparing $\gamma$-ray and CR induced showers. The Figure presents the distribution of the particle densities detected by the AS array ($\sum \rho$) against the total signal in the MD ($\sum N_\mu$). The distribution for the case of $\gamma$-ray showers is shown on the left panel of the Figure and the distribution for CR is shown on the right panel. In the analysis, the lower limit on the number of muons is defined as 0.1 for all the MD cells, and events with $\sum N_\mu < 0.1$ are piled up at around $\sum N_\mu = 0.01$. Gamma-ray events are observed to be muon-poor, while CR events are muon-rich and therefore; $\sum N_\mu$ is useful to discriminate between them.

For 100 TeV $\gamma$-rays, the angular and energy resolutions are estimated at about 0.2° and 25 %, respectively. The hadron rejection power of MDs is more than 99.9 % at 100 TeV, while keeping about 90 % of the $\gamma$-ray efficiency. Figure 4 shows the sensitivity curve of ALPAQUITA,

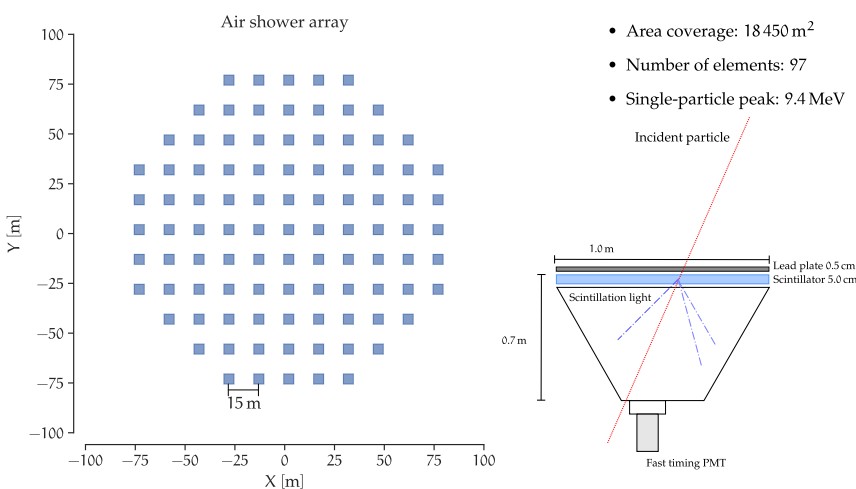

Figure 1: Schematic diagram of the ALPAQUITA array and principle of function

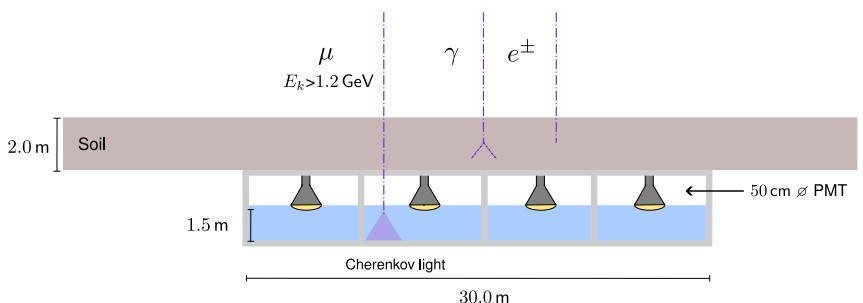

Figure 2: Schematic diagram of one underground MD pool in the ALPACA experiment and principle of function. Figure taken from [10].

along the energy spectra of 3 $\gamma$-ray sources that could be detected within one year of observation. The detection of gamma rays beyond 100 TeV is possible for HESS J1702-420A if the spectrum extends without cutoff [9]. The blue solid line represents the energy spectrum of Crab Nebula as reference. A more detailed study including the full array is presented in [9].

## 3  Construction plan and current status

In 2022 we installed an array of 97 detectors, which comprises the ALPAQUITA (little-ALPACA). This initial configuration is shown in the left panel of Figure 5. After the initial operation and maintenance, ALPAQUITA has been stably operating since April 2023. By the end of 2024, the construction of the first MD is about to start, and we will cover the surface with additional 60 detectors (configuration shown of the right panel of 4). Considering this, the gamma-ray sensitive operation of ALPAQUITA SD+MD will start in 2025. Figure 6 shows a panoramic view of the ALPAQUITA in 2023.

The air shower trigger signal of ALPAQUITA/ALPACA is issued when any four fold coincidence appears in the detectors, each of which records more than $\sim 0.6$ particles within a coincident width of 600 ns. Under these conditions, the trigger rate of ALPAQUITA is approximately 280 Hz. The modal energy of observed cosmic-ray air showers is estimated to be approximately 7 TeV by MC simulation [11].

Events are reconstructed using the relative timings and particle densities from the scintillation detectors. The timing information is corrected using the cable length fro each detector,

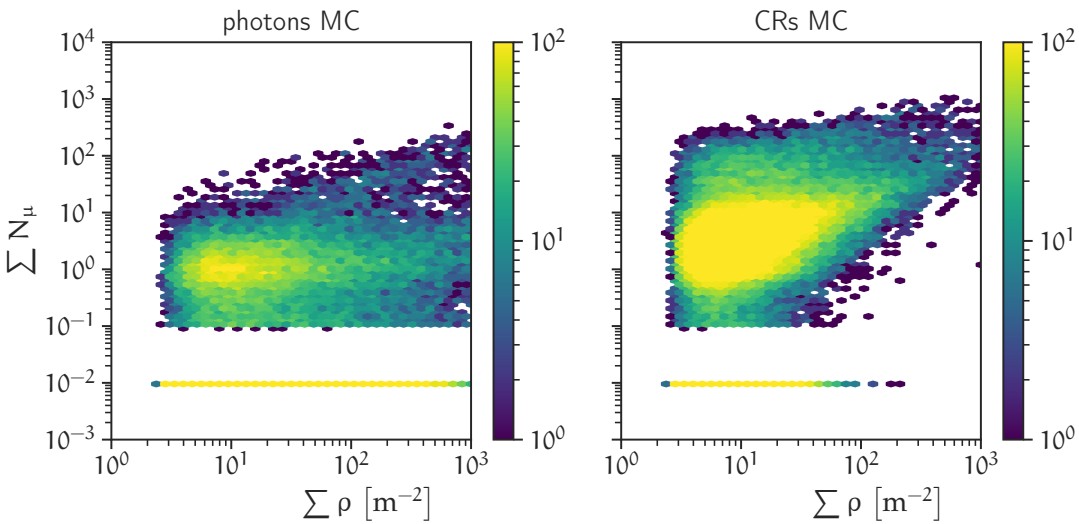

Figure 3: Distribution of the $\sum N_\mu$ against particle density $\sum \rho$ for different types of showers. Distribution in the left corresponds to $\gamma$-ray events and the one on the right is for CR. Events with $m_{k=0} < 0.1$ are accumulated at the bin $10^{-2}$

which is measured every 20 min and helps reduce the temperature effects in the determination of the arrival direction. The PMT transit times is also required to measure relative timings precisely, therefore we characterize this by measuring cosmic muons simultaneously each detector in coincidence with a small reference scintillation detector placed on top. At each detector, the particle density is obtained from the PMT output charge divided by the charge of the single particle peak [11]. The event reconstruction routine first determines the AS core location using hit detector positions weighted by the detected particle densities, then the timings in the AS front are fitted with a conical shape located at the core location to determine an arrival direction.

To test the angular resolution of the array we applied the *even-odd* analysis. In this, first we divide the detectors into 2 sub arrays and determined independently the arrival directions of the same shower with the two sub arrays. The distribution of the angular difference of two directions is shown in Figure 7. The blue crosses histogram shows the result of the experimental data, while the green hatched histogram shows the result of MC simulation, assuming the cosmic-ray spectrum and mass composition. Both distributions agree well, having a median values of 1.95°0.01 and 1.76°0.03 for experiment and MC, respectively. The median values are related to the angular resolution of the array but degraded by two factors. One is the reduction of statistics by splitting the data in half, and the other is the differentiation of the two reconstructed directions. By considering each term contributes by a factor $\sqrt{2}$, the angular resolution of the ALPAQUITA array is estimated to be $\sim 1°$ .

A more accurate measurement of angular resolution can be obtained by observing the shadow of the Moon in the cosmic ray flux. The first hypothesis of the Sun and the Moon (an apparent size of 0.5°) casting a shadow in high energy cosmic rays was presented in [13]. Taking this into account, the Tibet AS array successfully detected the moon shadow and used it to monitor the pointing accuracy and angular resolution [14]. Figure 8 shows the Moon shadow observed by the ALPAQUITA AS array for 310 live days from April 2023 to May 2024. The peak deficit position of the shadow is observed to shift westward from the apparent Moon position, which is an expected result due to the influence of the geomagnetic field. The statistical significance of the peak deficit position is estimated to be $-8\sigma$. The angular resolution

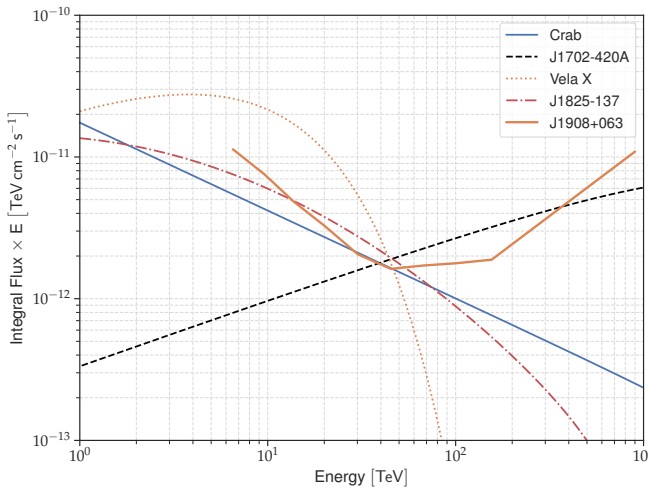

Figure 4: Sensitivity curve of ALPAQUITA with the $\gamma$-ray spectra of 3 sources in the field view that can be detected within one year of observation.

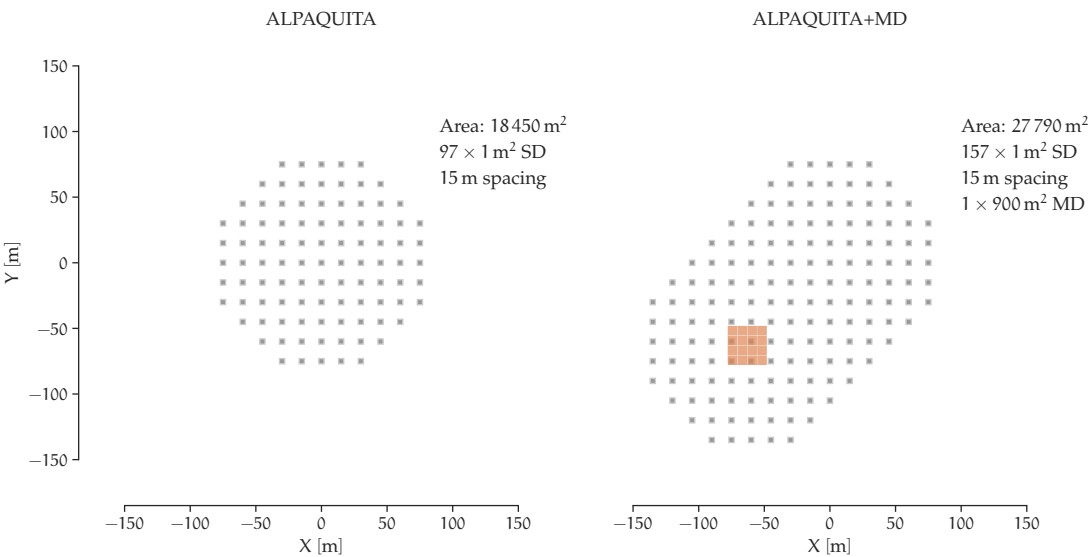

Figure 5: Construction plan of the ALPACA experiment. Left panel: layout of the ALPAQUITA array. Right panel: After the construction of the first MD unit, the surface array will be extended as enclosed by the solid-line octagon and will be operated as ALPAQUITA + MD. Full coverage array ALPACA will be eventually achieved

estimated from the Moon shadow observation is approximately 1° which is consistent with the MC simulation.

# 4  Conclusion

>100 TeV gamma-ray observation is crucial to search cosmic-ray accelerators called PeVatrons. Tibet AS$\gamma$, HAWC, LHAASO Collaboration has pioneered sub-PeV energy gamma-ray astronomy in the Northern Hemisphere. The ALPACA experiment is a project aiming to search for sub-PeV gamma-ray sources in the Southern Hemisphere for the first time using a new air

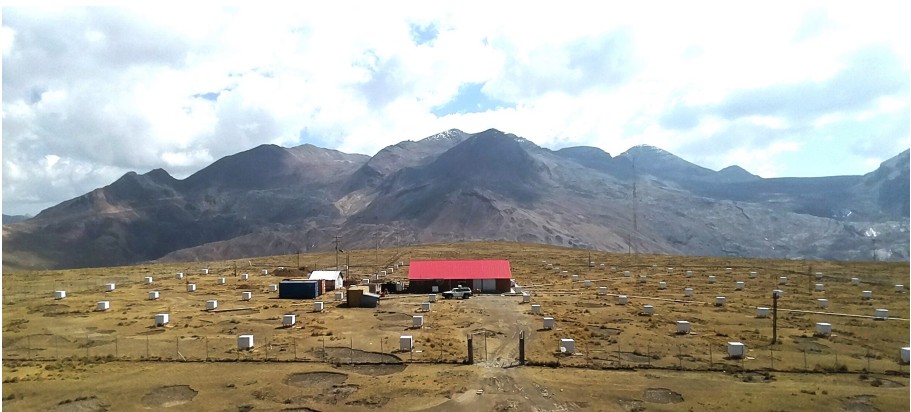

Figure 6: Panoramic view of the ALPAQUITA AS array located at the Chacaltaya plateau in Bolivia (June 2023)

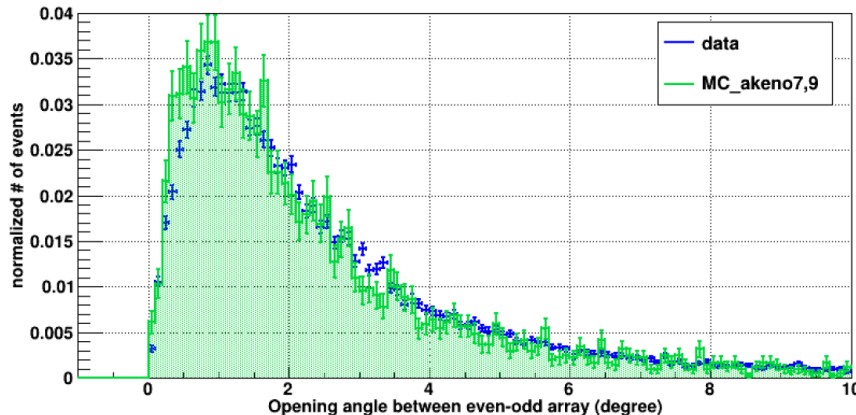

Figure 7: Distributions of the angular difference between two directions determined by the two sub arrays. Blue crosses shows the result of the experimental data while the green hatched histogram show the result of Monte Carlo simulation. Figure taken from [12].

shower array in Bolivia. The ALPAQUITA AS array has been successfully operated at Chacaltaya plateau in Bolivia, located at an altitude of 4740 m. With this array, we have detected the cosmic-ray Moon shadow with a statistical significance of $-8\sigma$. The angular resolution estimated based on the Moon shadow observation is 1.0°, which is consistent with results obtained by the MC simulation. By the end of 2024, we will start construction of an underground MD pool with and the extension of the AS. Furthermore, in 2025, we are planning to start construction of the full-scale ALPACA AS array and three additional underground MD pools. The ALPAQUITA and ALPACA experiments will help answering the long-standing problem, the origin of galactic cosmic rays, by opening a new gamma-ray window to the southern sky in the sub-PeV energy range.

## Acknowledgments

The ALPACA project is supported by the Japan Society for the Promotion of Science (JSPS) through Grants-in-Aid for Scientific Research (A) 24H00220, Scientific Research (B) 19H01922, Scienti fic Research (B) 20H01920, Scientific Research (S) 20H05640, Scientific Research (B) 20H01234, Scientific Research (C) 22K03660, and Specially Promoted Research 22H04912,

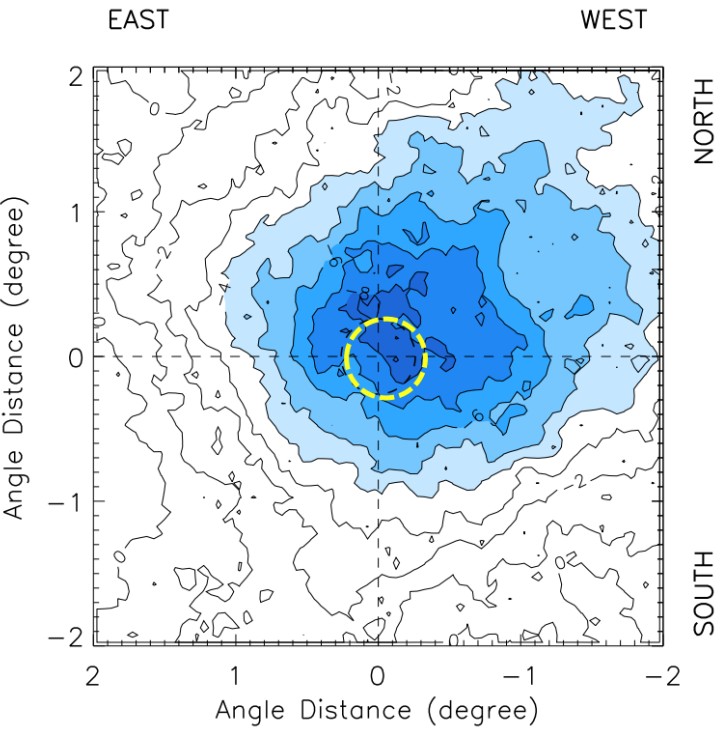

Figure 8: 2D significance map of the Moon shadow observed by the ALPAQUITA air shower array for 310 live days. The map is smoothed by a circle of an angular radius of 1.0°. The maximum significance is estimated to be $-8\sigma$. A dashed circle in the center of this map shows apparent size of the Moon.

the LeoAtrox super-computer located at the facilities of the Centro de Análisis de Datos (CADS), CGSAIT, Universidad de Guadalajara, México, and by the joint research program of the Institute for Cosmic Ray Research (ICRR), The University of Tokyo. Y. Katayose is also supported by JSPS Open Partnership joint Research projects F2018, F2019. K. Kawata is supported by the Toray Science Foundation. E de la Fuente thanks the Inter–University Research Programme of the Institute for Cosmic Ray Research (ICRR), University of Tokyo, Grant 2024i–F–05. I. Toledano-Juarez acknowledges support from CONACyT, México; grant 754851.

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
