# Peer review of "A new air shower array in the Southern Hemisphere looking for the origins of Cosmic rays: the ALPACA experiment"

_SciPost Physics Proceedings_

## Round 1 · Referee Report · Anonymous (Referee 1) · 2025-9-3

Strengths

The strengths are:

1) Above 100 TeV gamma ray measurements in the southern hemisphere. 2) Detection of moon shadow at high significance that provides calibration of various parameters.

Weaknesses

The paper has the following weaknesses. However, this is fine considering this as a journal proceeding.

  1. The language writing needs improvement, and the abbreviations, consistency.
  2. Limited comparison of results with other major experiments.

Report

Yes, this fits with the journal's scope.

Requested changes

  1. muon detector (MD) is already abbreviated in the Introduction. No need to repeat in the following sections.
  2. Figure 8: Can you add the contour bar to understand the variations?
  3. Is the angular resolution extracted from the fit? If so, then provide the parameters with error.
  4. At the end of the Introduction: A typo is found 'planed'.
  5. In Conclusion, It doesn't appear nice to the paragraph starts with "">100 TeV"

Recommendation

Ask for minor revision

---

## Editorial Decision

resubmitted